

# Advertisement design in dynamic interactive scenarios using DeepFM and long short-term memory (LSTM)

Lingling Zeng[1] and Muhammad Asif[2]

[1] Art College, Xinxiang Engineering College, Xinxiang, Henan, China
[2] National Textile University, Faisalabad, Pakistan

## ABSTRACT

This article addresses the evolving landscape of data advertising within network-based new media, seeking to mitigate the accuracy limitations prevalent in traditional film and television advertising evaluations. To overcome these challenges, a novel data-driven nonlinear dynamic neural network planning approach is proposed. Its primary objective is to augment the real-time evaluation precision and accuracy of film and television advertising in the dynamic interactive realm of network media. The methodology primarily revolves around formulating a design model for visual advertising in film and television, customized for the dynamic interactive milieu of network media. Leveraging DeepFM+long short-term memory (LSTM) modules in deep learning neural networks, the article embarks on constructing a comprehensive information statistics and data interest model derived from two public datasets. It further engages in feature engineering for visual advertising, crafting self-learning association rules that guide the data-driven design process and system flow. The article concludes by benchmarking the proposed visual neural network model against other models, using F1 and root mean square error (RMSE) metrics for evaluation. The findings affirm that the proposed model, capable of handling dynamic interactions among images, visual text, and more, excels in capturing nonlinear and feature-mining aspects. It exhibits commendable robustness and generalization capabilities within various contexts.

Corresponding author
Lingling Zeng,
hippo0020@hotmail.com

## INTRODUCTION

The way that advertising is done has evolved significantly with the rise of the Internet and big data technology. Digital radio, mobile messages, mobile TV, and other media are used to deliver and distribute advertisements (*Damaševičius & Zailskaite-Jakšte, 2022*). Combining big data analysis and information fusion scheduling techniques is important to assess the efficacy of correct advertising delivery, create an adaptive evaluation model for its performance, and increase the efficiency of accurate advertising delivery. The effectiveness optimization evaluation of advertising precision delivery is completed using a combination of information fusion and big data statistical analysis methodologies (*Zhou et al., 2021*).

The creation of the pertinent advertising precision delivery algorithm and method research for efficiency evaluation has drawn a lot of interest.

There have been many suggested machine learning algorithms that can automatically merge features. For example, a factor decomposer (FM) automatically integrates second-order features using the inner product of feature potential vectors (*Hu, 2022*). The field-aware factorization machine (FFM), which is based on the factorization machine, proposes the idea with the area and differentiates feature connections from various areas, which can enhance modeling feature interaction. Deep learning (DL) applications have shown considerable excellent recently in the areas of speech recognition, computer vision, and natural language processing (*Su & Zhao, 2021*). Because deep neural networks have high autonomous representational capabilities, they can be used to forecast click-through rate (CTR), which is a very promising application of deep learning technology. Deep neural network-based models have been widely proposed and have produced accurate prediction results (*Wang et al., 2022*; *Guo et al., 2017*). These DL models now have two improved parts. These suggested models of approaches take advantage of potent feature information, including deeper, wider networks and converters, to better understand higher-order interaction, which is crucial for enhancing CTR achievements in order to create dense low-dimensional vectors that are better suited for the input of the modeling approach, embedded skills are first utilized to close high-dimensional sparse feature vectors.

To increase the effectiveness of the recommendation system, a model to infer users' interests from their activity must be developed because people rarely express all of their awareness of social networks in an active fashion. The majority of deep learning models, on the other hand, place more emphasis on capturing the interaction between features and other classifications than on accurately portraying user interests. Despite the fact that there are several excellent models (*Richardson, Dominowska & Ragno, 2007*; *Chen et al., 2011*; *Srivastava, Mansimov & Salakhutdinov, 2015*; *Chu, 2021*) that can capture users' interests from previous projects they loved, these techniques can only capture the shown of users' trueless interests. Yet, when we are aware of what consumers dislike, we can avoid imposing efforts on people who do not like them more successfully. The objects that users have previously engaged with must typically be used as input by the method to catch the interests of users (*Xiang, Zhang & Li, 2021*). The number of projects may be high and the model may be easily over-fit because the majority of projects have only occasionally interacted (*Szegedy et al., 2016*; *Zhang et al., 2018*; *Guo et al., 2020*; *Abdel-Hamid et al., 2014*).

In this study, a novel model is proposed which, under dynamic interaction, model updating will greatly reduce the risk of over-fitting. The main contributions are as follows:

- For these two benefits, the proposed neural network model cleverly combines DeepFM and long short-term memory (LSTM). The DeepFM component can independently mix the entire range of user and advertising attribute features. In order to overcome the drawback, the LSTM component is crucial for extracting user interest and temporal change from click behavior.

- We created a model that included both forward and reverse feedback from user input. Users are able to discern the user material in which they are genuinely interested in this way.
- We put the proposed model to the test on two actual datasets. The experimental findings show that our method considerably enhances click-through rate prediction's performance and generalizability.

## RELATED WORK

With the development of business informatization; the calculation of click-through rates is used in a variety of online video visual advertising contexts, including personalized advertising, friend and product recommendations, *etc*. New hit rate estimation models are therefore frequently put forth (*Zhu et al., 2020*). In general, the deep neural network model has replaced the classic machine learning model as the go-to model for click-through prediction, and the model's shallow structure has been replaced with a deep structure. We will thoroughly describe the related study of click-through behavior in this part.

Most of the time, it is possible to convert the hit rate prediction problem into the user of maximum interests with cross-entropy in goals. As a result, many conventional machine learning methods are applicable as divination methods (*Ke et al., 2017*). Such as the linear logistic regression methods linearly increase the entering area of application before compressing the value to a range between 0 and 1. Since there is no nonlinear modification of the original entering areas, automatic feature combining is not possible. It requires specialist skills to manually implement higher-order combination features.

The well-known factorization machine is suggested, which employs the inner co-conference of the areas vectors as the areas that interact and retain dealing with each area of input. The entering performance of CTR is significantly enhanced by factor decomposer, which can catch the interplay between FIFO information (*Feng et al., 2019*). Then, various alternate factorization-based models are put out. FFM facilitates the interaction between features in many domains by maintaining numerous feature vectors in several domains for a given feature. It has been successfully demonstrated in computer vision, natural language processing, and speech recognition that more social networks of applications can represent features.

Consequently, using deep learning to examine higher-order features in the context of CTR is quite interesting. The factor decomposer-supported neural network (BP) obtains the areas vector—the embedded vector of feature embedding—by using the pre-trained FM model and then connects the embedded vector to the feedforward neural network (*Zhang et al., 2018*; *Song et al., 2019*).

DeepFM is a machine learning model that combines the strengths of factorization machines (FM) and deep neural networks (DNN). The FM component of DeepFM is responsible for capturing the first- and second-order feature interactions, while the DNN component captures higher-order feature interactions. The FM component is based on logistic regression. It captures pairwise feature interactions, while the DNN component is a fully connected neural network that can capture more complex, higher-order feature interactions. The key innovation of DeepFM is the way it combines these two

components to jointly model low-order and higher-order feature interactions (*Feng et al., 2022*). Specifically, the FM component is used to model the low-order feature interactions and is fed into the DNN component as an input. The DNN component then interacts with the FM output to model higher-order feature interactions. This architecture allows the model to effectively capture both low-order and higher-order feature interactions, making it well-suited for a wide range of recommendation and prediction tasks (*Yu et al., 2019*).

Such as applications of the deep Internet of networks need to replicate the temporal evolution of interest in addition to extracting user interest. From the user's previous interactive project list, these models solely extract the user's active interest. Nonetheless, we believe that consumers' unfavorable interests are also crucial. The model can prevent proposing things that people don't like by knowing what they dislike. More items of data are used as complex user interest calculation data. Still, most of them only show up a few times, increasing the risk of over-fitting and making it difficult for the model to be generalized. As a result, using long and short-memory neural networks (LSTM) will address the issue that LSTM is a great recurrent neural network (RNN) variant, inherits the traits of the majority of RNN models, and addresses the Vanishing Gradient problem brought on by the gradual slowing down of gradient backpropagation process, which is something to take into consideration (*Hamdy, Shalaby & Sallam, 2020*).

## SYSTEM MODEL

The proposed model ingeniously merges the strengths of DeepFM and LSTM. Within this model, the DeepFM component showcases its prowess by autonomously blending a comprehensive array of user and advertising attribute features. This intrinsic capability ensures a holistic understanding of the diverse facets influencing user engagement. To address a potential limitation, the LSTM component becomes indispensable, offering a solution by adeptly extracting user interests and deciphering temporal changes derived from click behavior.

Suppose that the training data set (X, Y) is composed of n sample sequences. At present, for the sequence of each sample (x, y), where x is the fixed feature vector, and the m field and y are the values of binary units. The feature vector x in demand includes the user's feature attributes and sample attributes (such as gender, age, and occupation), film and television advertising features (such as advertising name, advertising introduction, and evaluation indicators), and context-aware information for mapping advertising (such as geographic information, user information, and marketing strategy). The binary value y {0,1}, which represents the above m field and the field in y, indicates that the user has incentive behavior and operation steps (where 0 and 1 represent whether or not the advertisement has been clicked). At the same time, there may be different types of field x in digital advertising, representing continuous and discrete features. In particular, continuous features and context-aware features are encoded by one-pot coding. The discrete features and contextual features need to be further perceived; for example, the comment context may be, click on the first item, and the second item is associated with the information advertisement, but not click on the third item, so the example is x [1,1,0]. Our goal is

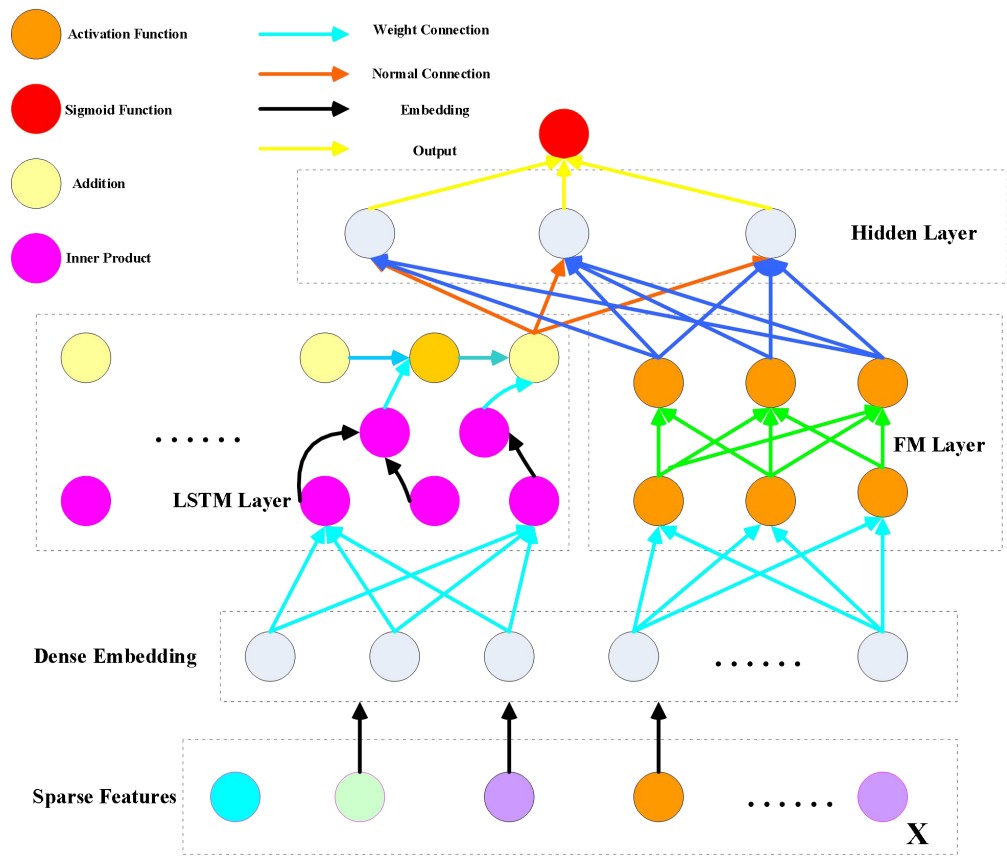

**Figure 1  DeepFM model architecture.**

to create a new CTR prediction model to better capture user interest by capturing user interest from the user's previous behavior sequence. Therefore, we propose a visual neural network model combining LSTM and DeepFM, and its architecture is shown in Fig. 1.

The relationship between factorization machines (FM) and LSTM parameters holds significant implications. While initializing FM embedding *via* pre-training might seem like a viable approach, it doesn't always guarantee precise results. This method can escalate computational complexity and hinder training efficiency while potentially neglecting low-order feature modeling within the model's architecture.

In response, LSTM introduces a product layer sandwiched between the first hidden layer and the embedded layer. This intermediary step aims to capture higher-order features. However, implementing this product layer necessitates a fully connected output between the product layer and the first hidden layer, which significantly raises computational complexity. Moreover, the inner product utilized here tends to be computationally expensive, while the outer product approximation method, while more efficient, can suffer from instability issues.

On the contrary, DeepFM takes a different approach by forgoing pre-training the hidden vector V with FM. Instead, this module undergoes simultaneous training alongside the

entire model, sharing feature embedding. Through component integration, the model generates a comprehensive output without relying on pre-training methods. This approach allows for a more streamlined training process and avoids the pitfalls associated with high computational complexity and instability, offering a promising alternative for optimizing the model's performance.

$$y = \text{sigmoid}\left(y_{FM} + y_{LSTM}\right) \tag{1}$$

The attention mechanism's factorizer model presents a promising avenue for optimizing modeling approaches. The original model, featuring a parallel structure, independently calculates each part and consolidates outcomes at the output layer. In a modification of this approach, the model incorporates the first and second terms of FM as input, creating a network model with a serial structure. This factorizer, operating as a supervised learning method, enhances linear reconstructed models by incorporating second-order finite element interactions.

However, challenges arise from the equal weighting of all feature interactions, as not all interactions contribute equally to the model's performance. The inclusion of irrelevant feature interactions may introduce noise, adversely impacting performance. To address this issue, an attention mechanism is integrated into the factorizer, allowing the model to discern the importance of different feature interactions. This enhancement, learned through neural attention networks, results in improved regression accuracy, a simplified model structure, and a reduction in the number of model parameters.

Leveraging the collaborative power of features often involves extending eigenvectors through the incorporation of feature products—an approach widely embraced to unlock deeper insights. Consider polynomial regression, an exemplar of this technique, adept at capturing the nuanced weights associated with intersecting features. However, a substantial hurdle emerges when dealing with sparse datasets where only a handful of cross-features are observable. This limitation renders the inference of parameters for unobserved cross-features nearly impossible.

Enter the attractor-attention mechanism, a promising solution that enriches the performance of FM by discerning the importance of feature interactions. This mechanism operates through a dual-layered architecture: the coupled interaction layer and the attention-based concentration layer. Here, the mechanism transforms n vectors into (n-1)/2 interacting vectors, each representing the product of two distinct vectors, effectively encoding their interactions.

The brilliance lies in this transformational process, where these interaction vectors serve as critical indicators of feature interplay. By encapsulating the intricate relationships between pairs of vectors, the mechanism optimizes the understanding of feature synergies despite the scarcity of observed data points. This innovative mechanism holds the promise of significantly improving the predictive capability of models like FM in scenarios marked by limited cross-feature observations.

Let X represent the set of non-zero eigenvalues in the eigenvector x. The output of the coupled interaction layer can be formalized as a set vector. To tackle the generalization problem, a multi-layer perceptron (MLP) is utilized to parameterize the attention score.

The attention force network takes the interaction vector of two features as input to encode their interaction information in the embedded space.

$$a'_{ij} = h^T \text{ReLU}\left(W\left(v_i \odot v_j\right) x_i x_j + b\right) \tag{2}$$

$$a_{ij} = \frac{\exp\left(a'_{ij}\right)}{\sum_{(i,j) \in R_x} \exp\left(a'_{ij}\right)} \tag{3}$$

where W, b, and h denote the model parameters, whereas t represents the size of the attention network's hidden layer, commonly referred to as the attention factor. Notably, the score is normalized by the softmax function, a prevalent approach in prior research. The attention-based network generates a K-dimensional vector that consolidates all feature interactions, which is subsequently mapped to the predicted score. The model for this branch can be formalized as follows:

$$\hat{y}_{\text{AtFM(X)}} = w_0 + \sum_{i=1}^{n} w_i x_i + p^T \sum_{i=1}^{n} \sum_{j=i+1}^{n} a_{ij}\left(v_i \odot v_j\right) x_i x_j \tag{4}$$

Considering that FM are directly improved from a data modeling perspective, they find applicability in various prediction tasks, including regression, classification, and sequencing. To customize the model learning for specific tasks, diverse objective functions should be employed. In the context of regression, where the target y(x) represents the actual value, the squared loss is commonly adopted as the objective function:

$$L_r = \sum_{x \in \tau} \left(\hat{y}_{\text{AtFM}}(x) - y(x)\right)^2 \tag{5}$$

where $\tau$ denotes the training set, and logarithmic losses can be leveraged for binary classification or recommendation tasks. Stochastic gradient descent is utilized to optimize the objective function.

Most of the attributes of the input samples fall under the category of traits. The one-hot vector representing a combination of category features has a high dimension and is highly sparse, making it unsuitable for direct input into neural networks. Therefore, it needs to be transformed into a low-dimensional dense vector, often referred to as an embedded vector (*Yang, Zheng & Xiao, 2022*). However, replacing the original feature with an embedded vector results in some loss of data. In general, there is little to no correlation between the traits of different categories. For instance, a user's age and gender are examples of natural categories describing the person or object, and age and sexual orientation are unrelated (*Khlifi, Boulila & Farah, 2023*).

## EXPERIMENTS AND ANALYSIS

By utilizing established metrics, the article not only substantiates the model's proficiency but also contributes valuable insights into its relative strengths and capabilities, thereby solidifying its standing as a promising and competitive solution in the realm of visual data

**Table 1 Experimental setup parameters.**

| Serial number | Environments | Configure |
|---|---|---|
| 1 | Language | Python |
| 2 | Tool kit | Numpy,sklearn |
| 3 | Software | Python3.7 jupyter,notebook, pandle |
| 4 | Hardware | Two laptops , |
| 5 | Operating System | Ubuntu 16.04 |

analysis. The proposed model is compared with language model (LM), FM, deep neural network (DNN) and collaborative filtering (CF).

Language model (LM): Learns the probability distribution of language, used to predict the next word or character given a text.

Factorization machine (FM): Handles sparse data, particularly suitable for high-dimensional, sparse features.

Deep neural network (DNN): A neural network composed of multiple layers used to learn complex representations of input data.

Collaborative filtering (CF): A recommendation system approach that predicts items or content a user might like based on the analysis of user behavior and preferences.

## Experimental settings

To avoid overfitting the model, we remove the listed nouns and unknown nouns from the ml-Lasts-small and ml-100k data sets of the University of Minnesota GroupLens recommendation algorithm, respectively. These data sets contain roughly 100,000 evaluation records. For training and verification, we divide these two data sets at random into two groups. Next, we examine how the vision-based neural network model performs on the two public data sets mentioned above. Table 1 describes the experimental setting and tools:

## Model training

The lifting curve for the visual neural network for dynamic interaction is computed using the two open data sets, as seen in Figs. 2 and 3. The sampling points in this picture correspond to 10 comparable nearby k values from 1 to 100, which are these two indexes. "Treat1" refers to the DeepFM and LSTM modified method, while "Treat2" refers to the collaborative filtering closest neighbor operator. The updated method has a smaller depth and a greater degree of improvement than the classic approach at the same k point for the two common data sets. This indicates that the modified algorithm can achieve a bigger improvement effect with fewer recommendations.

The article conducts a comprehensive analysis of the recommendation effect's stability and enhancement potential through a meticulous examination of evaluation indicators. It delves into the intricate relationship between the subject's work characteristic curve (ROC) and the promotion degree curve, deciphering their significance in gauging the efficiency of recommendations. The ROC's abscissa represents the false positive rate (FPR), while the ordinate portrays the true rate (TPR), encapsulating recall rate and sensitivity. Each point

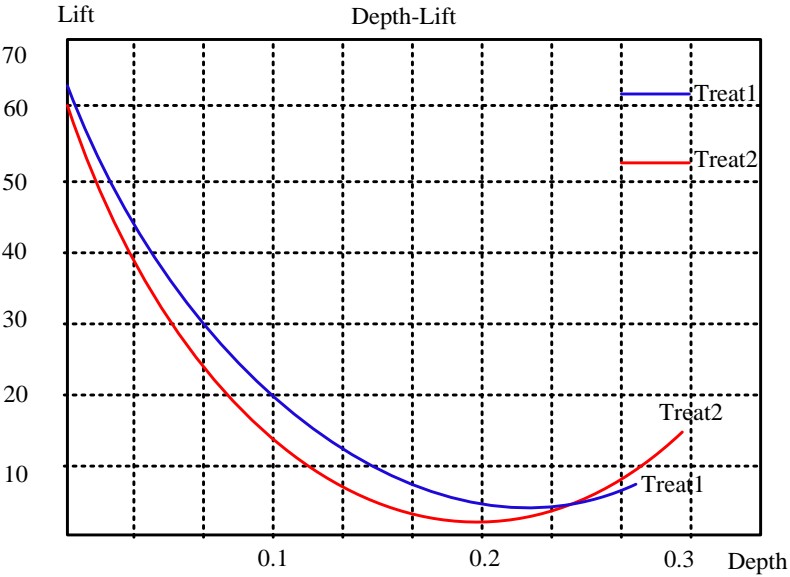

**Figure 2** **Lifting curve of ml-latest-small dataset.**

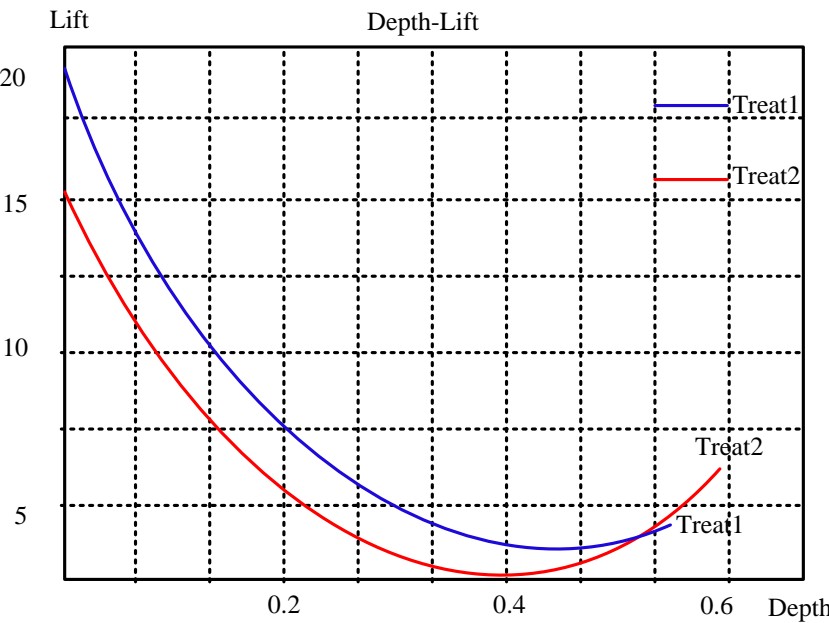

**Figure 3** **Lifting curve of ml-100k dataset.**

on this curve signifies the nuanced performance contrast between FPR and TPR across various threshold signal stimulations.

This assessment serves the critical purpose of minimizing misclassification of negative samples as positives, thereby ensuring a more accurate identification of positive individuals.

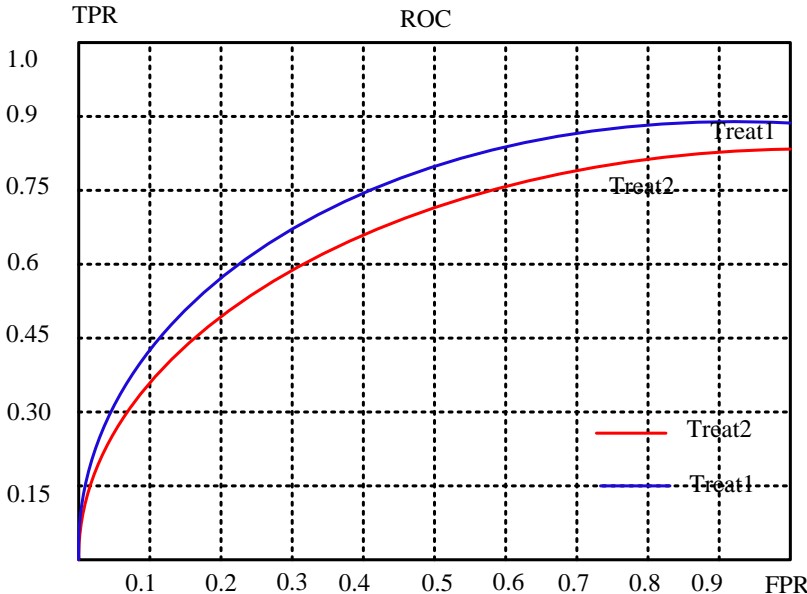

**Figure 4  Subject characteristic curve of ml-latest-small dataset.**

Ideally, arranging samples based on score rankings optimizes this process. A favorable scenario exhibits TPR progressively ascending from 0 to 1, while FPR steadfastly maintains at 0. Conversely, the worst-case scenario manifests as a linear ROC from 0 to 1, signifying equality between TPR and FPR within each partition.

## Dataset

The article provides a comparative analysis between two distinct algorithms through the presentation of Figs. 4 and 5. These figures delineate the performance of the algorithms across separate datasets. This comparative exploration contributes to an enhanced understanding of the efficacy of the algorithms within these specific datasets.

To establish a standardized benchmark in the algorithmic solution process, the recommended length, denoted as 'k,' is divided into ten equal segments. This tailored approach aligns with the evaluation system and indicators detailed in the preceding experimental analysis. Each segment corresponds to a specific evaluation index value for the suggested algorithm, facilitating a meticulous comparison between the outcomes of two distinct models: the conventional collaborative filtering closest neighbor algorithm and the curiosity-guided group-modified recommendation algorithm.

Through this systematic evaluation, distinct scenarios are scrutinized to discern the performance differentials between the two models. The results paint a compelling picture: the novel algorithm showcases superior performance under select circumstances. Notably, it exhibits heightened robustness and stability in comparison to the conventional collaborative filtering technique.

This nuanced comparison unveils the strengths of the curiosity-guided group-modified recommendation algorithm, illuminating its efficacy in scenarios where conventional

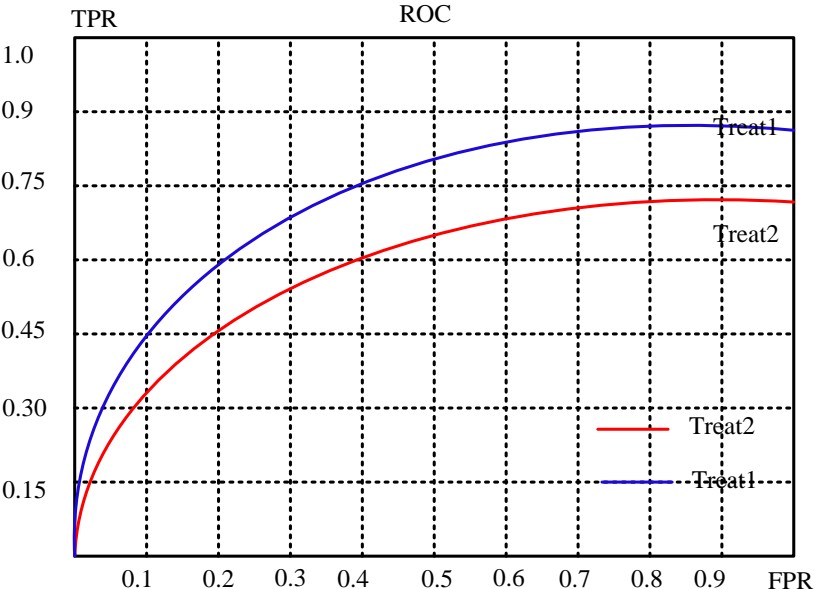

**Figure 5** Subject characteristic curve of ml-l00k dataset.

**Table 2** Comparison of different models on ml-100k dataset.

| Models | F1 | RMSE |
|---|---|---|
| LM | 0.651 | 0.353 |
| FM | 0.668 | 0.378 |
| DNN | 0.653 | 0.354 |
| CF | 0.643 | 0.346 |
| **Deep+LSTM** | **0.674** | **0.368** |

Notes.
   The evaluation indicators and the optimal model are shown in bold.

methods falter. By outperforming in specific conditions and demonstrating heightened stability, this novel algorithm emerges as a promising contender, potentially reshaping recommendation systems by offering more reliable and adaptable solutions.

## Model comparison

The insights gleaned from Tables 2 and 3 regarding CTR prediction performance across various models in the specified public datasets reveal crucial factors dictating efficacy:

Emphasis on automatic higher-order feature combining: A pivotal finding underscores the significance of automatic higher-order feature combining in enhancing CTR effectiveness. The performance gap between the LR model and deep neural network models capable of autonomously amalgamating higher-order features highlights this. Notably, the DeepFM+LSTM model showcases superior predictive capabilities over LR, emphasizing its capacity for automatic higher-order feature synthesis.

**Table 3  Comparison of different models on ml-latest-small dataset.**

| Models | F1 | RMSE |
|---|---|---|
| LM | 0.642 | 0.346 |
| FM | 0.657 | 0.362 |
| DNN | 0.632 | 0.354 |
| CF | 0.643 | 0.351 |
| **Deep+LSTM** | **0.667** | **0.358** |

Notes.
The evaluation indicators and the optimal model are shown in bold.

Performance advantages of DeepFM+LSTM: The DeepFM+LSTM model exhibits substantial improvements in predictive metrics within both the ml-100k and ml-latest-small datasets. Specifically, F1 and RMSE metrics witnessed significant enhancements. In the ml-100k dataset, the DeepFM+LSTM model displayed an increase of 3.53% and 4.24%, respectively, while in the ml-latest-small dataset, improvements of 3.13% and 3.46%, respectively, were observed. These results underline DeepFM+LSTM's proficiency in achieving a balance between accuracy and recall, albeit with a marginal trade-off in accuracy.

The findings affirm the pivotal role of automatic higher-order feature synthesis in CTR prediction effectiveness, highlighting the DeepFM+LSTM model's superiority in reconciling accuracy and recall metrics while underscoring the trade-off encountered in precision. This emphasizes the need for a balanced approach, considering the intricate interplay between accuracy, recall, and feature synthesis mechanisms to optimize CTR prediction models.

The figures, specifically Figs. 6 and 7, underscore the profound impact of comprehending user click behavior on enhancing the accuracy of click rate predictions. This observation stems from the superior performance of the DNN model, which excels in prediction compared to other deep neural network models incapable of effectively capturing sequence features. In the ml-100k dataset, the DeepFM+LSTM model exhibits a noteworthy improvement, elevating F1 and RMSE metrics by 1.04% and 1.23%, respectively, in contrast to the FM model. Impressively, within the ml-latest-small dataset, this enhancement surged by 2.11% and 4.53%, respectively.

Moreover, the study reveals a correlation between the depth of user interest description and the predictive capacity of click-through models. The LSTM model's superior performance over DNN underscores this connection, further emphasizing the potency of LSTM, particularly when employing gated neural networks vis-à-vis DRNN for sequence feature modeling. In the ml-100k dataset, the DeepFM+LSTM model exhibits substantial growth, witnessing a surge of 4.07% and 44.08% in F1 and RMSE metrics correspondingly. Similarly, in the ml-latest-small dataset, there's an increase of 3.14% in F1 and 1.12% in RMSE metrics.

The DeepFM+LSTM model exhibits advantages in two key aspects: interaction modeling and temporal modeling.

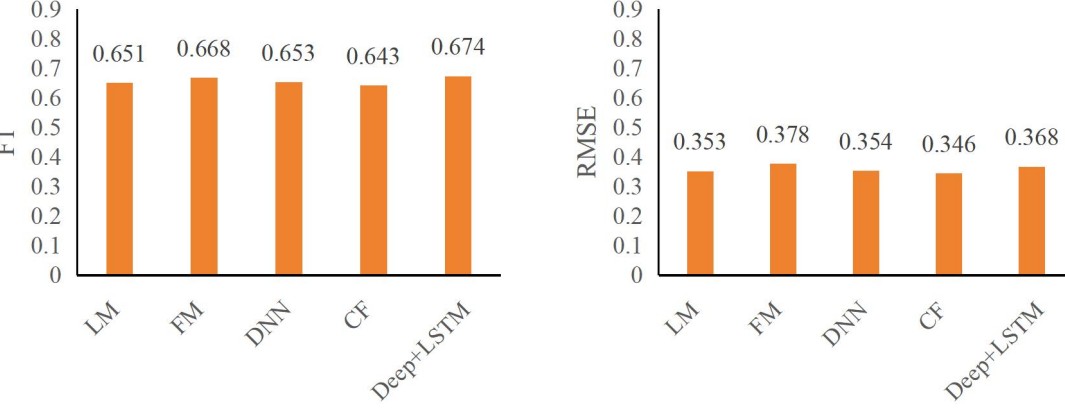

**Figure 6**  **F1 and RMSE comparison among different models on ml-100k dataset.**

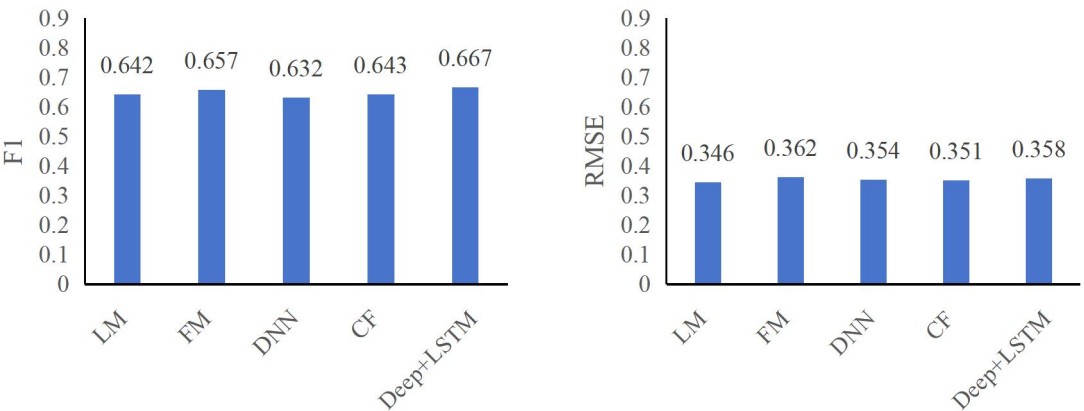

**Figure 7**  **F1 and RMSE comparison among different models on ml-latest-small dataset.**

Firstly, by integrating DeepFM, the model excels in capturing the interaction dynamics between users and items. DeepFM combines factorization machines with deep neural networks, effectively learning high-order interactive relationships between features. This enables the model to more accurately understand users' latent interests and preferences, enhancing the precision of recommendations. Factorization machines contribute to handling sparse data, while deep neural networks capture more complex nonlinear relationships, making the model more expressive.

Secondly, the introduction of LSTM for temporal modeling, especially in handling time-series data of user behavior, is a notable strength. Recommender systems often face the challenge of temporal evolution in user interests and behaviors, and LSTM efficiently captures long-term dependencies within sequence data. This allows the model to better comprehend users' historical behavior and make more precise predictions about future interests. By considering temporal factors, the DeepFM+LSTM model adapts more

effectively to the dynamic changes in user behavior, increasing the personalization level of the recommender system.

However, it is essential to note that while DeepFM+LSTM demonstrates commendable accuracy and recall harmony, its utilization heightens both the model's complexity and accuracy up to a certain threshold. This trade-off suggests a balance between model intricacy, precision, and recall, advocating for judicious consideration of these factors in the pursuit of optimal predictive performance.

## CONCLUSIONS

Automatic feature combination is a pivotal technology in CTR prediction, facilitating the exploration of deeper correlations between features. While numerous high-quality deep learning models have been proposed for this purpose, the primary objective of this article is not to enhance feature combinations but to create them using DeepFM. To captivate consumer interest, several in-depth learning models focusing on user history interaction have been introduced. The Deep Interest Network, for example, employs an attention mechanism to gauge a user's interest in the current project based on their previous purchases. In this study, we cleverly represent a user's prior interactions with a sequence of 1s and 0s, model the sequence using a cyclic neural network, and subsequently deduce the user's interest and development. This approach effectively mitigates the risk of overfitting, thereby expediting model training and updates. When designing the response model structure, careful consideration is given to the model's generalization ability and various evaluation metrics, including mainstream indicators such as area under the curve, receiver operating characteristic, F1, recall, and error analysis for accuracy. In future research, we aim to further explore user and item information based on quantitative evaluation and analysis of film and television works. This may involve leveraging techniques such as the Boltzmann machine and random forest algorithm to design corresponding model advertising recommendation algorithms. These endeavors provide additional avenues for research in the evaluation and analysis of advertising design.

### Funding
The authors received no funding for this work.

### Competing Interests
Muhammad Asif is an Academic Editor for PeerJ.

### Author Contributions
- Lingling Zeng conceived and designed the experiments, analyzed the data, prepared figures and/or tables, and approved the final draft.
- Muhammad Asif performed the experiments, analyzed the data, performed the computation work, authored or reviewed drafts of the article, and approved the final draft.

## Data Availability

The raw data is available at Zenodo: Anonymous. (2023). AdFlush Dataset [Data set]. Zenodo. https://doi.org/10.5281/zenodo.10039834.

The code is available in the Supplemental File.

## Supplemental Information

Supplemental information for this article can be found online at http://dx.doi.org/10.7717/peerj-cs.1937#supplemental-information.

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
