# Peer review of "Advertisement design in dynamic interactive scenarios using DeepFM and long short-term memory (LSTM)"

_PeerJ Computer Science, doi:10.7717/peerj-cs.1937_

## Round 0.1 · original submission · Major Revisions

Based on the reviewers’ comments, you may resubmit the revised manuscript for further consideration. Please consider the reviewers’ comments carefully and submit a list of responses to the comments along with the revised manuscript.

**Language Note:** PeerJ staff have identified that the English language needs to be improved. When you prepare your next revision, please either (i) have a colleague who is proficient in English and familiar with the subject matter review your manuscript, or (ii) contact a professional editing service to review your manuscript. PeerJ can provide language editing services - you can contact us at copyediting@peerj.com for pricing (be sure to provide your manuscript number and title). – PeerJ Staff

Reviewer 1 ·

Basic reporting

The approach of this paper focuses on developing a design model for film and television visual advertising, customized for the dynamic interactive environment of online media. Using the DeepFM+LSTM module in a deep learning neural network, this paper begins to build a comprehensive information statistics and data interest model derived from two common datasets. The results confirm that the proposed model is capable of handling dynamic interactions between images and visual text, and performs well in capturing nonlinearity and feature mining. It shows commendable robustness and generalization in different environments. However, this article still needs to improve the following content:
a. The keywords of the abstract may need to be re-selected according to the content of the article;
b. Each section should be preceded by a relevant paragraph explaining the main content of the chapter in order to improve readability;

Experimental design

c. The description and explanation of the formula need to pay attention to not only the format should be unified, but also the matching degree with the model and the logic of the description;
d. In section 3, the nonlinear dynamic neural network is proposed for the new data. How is the nonlinearity realized and what is its function;
e. The various evaluation models used in the experiment should be introduced in section 4.2, such as listing formulas, etc;
f. There seems to be a processing of data length in section 4.2, and I would suggest that the introduction to the data set and the data preprocessing be described separately;

Validity of the findings

g. As for the experimental part, especially the experimental results in Fig. 6 and Fig. 7, I would suggest the author to highlight the advantages of this paper according to the results of model comparison and performance comparison;
Some of the references cited in the article are not the latest, and articles from excellent journals in recent years should be added.

Reviewer 2 ·

Basic reporting

This paper studies the development prospect of data advertising in web-based new media, and proposes a new data-driven nonlinear dynamic neural network planning method. Later in the research, the proposed vision neural network model is compared with other models, and F1 and RMSE indexes are used to evaluate the model, and the performance is very good. However, there are some shortcomings that need to be addressed in detail.

Experimental design

Overall the experimental design is appropriate but needs some inputs as follows:

1. The description of experimental results in the part is too small to support the innovation. I will suggest adding some experimental results of specific value types

2. Due to the lack of experimental comparison in the experimental part, we can consider adding ablation experiments to enhance the experimental results and highlight the innovation

Validity of the findings

1. The content of the related works, the logical connection between the various sections of the content needs to be strengthened for improved readablity.


2. The optimization method of attention mechanism is used in the part of model design. What problems does this solve and what innovations does it provide

3. The third part is about the data features mentioned in the last paragraph, how is the paper selected and processed

Additional comments

The conclusion part does not have too much summary of the main content of the article, and reads a little like part of the abstract. This needs to be revisited and extended.

From the perspective of the overall text polishing of the article, there may be some shortcomings, and it is necessary to re-strengthen the language and polish it

The contribution at the end of the introduction section is not described in a fixed format, and the author should make relevant changes and adjustments;

---

## Round 0.2 · accepted · Accept

Congratulations, the reviewers are satisfied with the revised version and have recommended the acceptance decision.

Reviewer 1 ·

Basic reporting

The you for making effort in addressing all the raised concerns.

Experimental design

no comment

Validity of the findings

no comment

Additional comments

no comment

Reviewer 2 ·

Basic reporting

The authors have incorporated the corrections/ enhancements suggested earlier. The readability, flow and practicality of the research has improved.

Experimental design

Experimental design and its execution has been extended and improvements are obvious.

Validity of the findings

Discussions and conclusions have been extended. Discussion on the research conclusion and its findings has also been improved.

Additional comments

Suggested: Accept.